# Recovery and partial isolation of α-mangostin from mangosteen pericarpsvia sequential extraction and precipitation

**Moh Moh Han[1], Preuk Tangpromphan[2], Amaraporn Kaewchada[3], Attasak Jaree[2]***

**1** Department of Chemical Engineering, Faculty of Engineering, Kasetsart University, Bangkok, Thailand, **2** Center for High-Value Products from Bioresources, Department of Chemical Engineering, Faculty of Engineering, Kasetsart University, Bangkok, Thailand, **3** Department of Agro-Industrial, Food, and Environmental Technology, King Mongkut's University of Technology North Bangkok, Bangkok, Thailand

* fengasj@ku.ac.th

**Data Availability Statement:** All relevant data are within the Supporting information file.

## Abstract

This study introduced an innovative sequential extraction methodology designed for the efficient recovery of alpha-mangostin (α-M) from mangosteen pericarps. Alpha-mangostin, renowned for its pharmacological properties including anti-inflammatory, anti-cancer, and anti-bacterial effects, has garnered significant attention across diverse industries. The proposed method of sequential extraction achieved 73% recovery and a yield of 46.75 mg/g based on the weight/weight percentage of the mass of α-M extracted from the sequence and the mass of raw material. Furthermore, the purity of the dried product was 67.9%. The sequence solvent extraction system, comprising water, hexane, and acetonitrile, plays a pivotal role in enhancing the efficacy of the extraction process. Notably, this methodology offers a cost-effective alternative to conventional extraction methods. It reduces the need for complex equipment and processes, positioning it as a resource-efficient extraction technique in comparison to existing methodologies. This novel sequential extraction method presents a promising avenue for the economical and sustainable recovery of alpha-mangostin (α-M) from pericarps.

## 1. Introduction

Mangosteen (Garcinia mangostana L.), often hailed as the "queen of fruits" due to its distinctive floral appearance, is a commercially significant crop in Thailand and Southeast Asia. While India leads global production, Thailand is a major producer in the region [1]. The fruit comprises 83% pericarps, 15% pulp, and 2% seeds [2]. Although the pericarps have limited initial market value, they possess substantial potential due to their rich content of bioactive compounds, particularly xanthones [3]. These compounds exhibit a wide range of health benefits, including antioxidant and anticancer properties, driving demand across medical, food, cosmetic, and oral care industries [4, 5].

Mangosteen pericarps contain over 68 types of xanthones, with 69.01% of alpha-mangostin (α-M) and 17.86% of gamma-mangostin (γ-M) being the predominant compounds [6].

**Funding:** This work was supported by the Faculty of Engineering, Kasetsart University. The funder had no role in study design, data collection and analysis, decision to publish, or preparation of the manuscript.

**Competing interests:** The authors have declared that no competing interests exist.

Alpha-mangostin, a hydrophobic polyphenol, is primarily responsible for the fruit's antioxidant capacity, highlighting the importance of purification processes to isolate this valuable compound [7]. Bioactive compounds can be extracted from mangosteen pericarp through solid-liquid extraction. While both fresh and dried pericarps are viable starting materials, the latter is generally preferred to minimize microbial contamination [8]. Factors affecting solid-liquid extraction include solvent and solid properties as well as the operating conditions such as extraction time, and temperature [9]. High temperatures can degrade bioactive compounds despite influencing solubility and diffusion rates. Various methods exist to extract bioactive xanthones from mangosteen pericarp. Conventional methods such as solvent extraction [10], maceration [11–13] and Soxhlet extraction [14] require substantial solvent volumes and prolonged durations. Due to these drawbacks, contemporary methods have been explored to retrieve phenolic and bioactive compounds. Microwave extraction with ethyl acetate and ultrasonic extraction with ethanol have shown promising results in extracting xanthones, notably α-M. The extraction yield of α-M and γ-M using methylene chloride as solvent was reported as 33.24 mg/g dried mangosteen (DM) and 8.61 mg/g of DM, respectively [6]. Ghasemzadeh et al. used microwave extraction with 72.40% (v/v) ethyl acetate, yielding 120.68 mg of xanthones/g of mangosteen pericarp (dry matter) [15]. Ultrasonic extraction with 95% (v/v) ethanol at 30 kHz and room temperature extracted α-mangostin and 8-desoxygartanin with yields of 47.82 ± 3.76 mg/g and 1.43 ± 0.30 mg/g (dry basis), respectively [16]. Supercritical water extraction was tested at 120–180°C and 1 to 5 MPa. The optimal xanthone yield was 34 mg/g of the sample at 180°C and 3 MPa, with a reaction time of 150 min [17]. Advanced techniques like supercritical carbon dioxide (SC–CO$_2$) [18, 19], ultrasonic bath [9], and microwave-assisted extraction (MAE) [15] show significant α-mangostin extraction efficiency but are costly and require specialized equipment.

The quality of extracted bioactive compounds depends on an effective extraction method, with a clear chromatographic peak indicating purity. Ensuring purity is crucial to prevent side effects in pharmaceutical or medical applications, as chemical variations can lead to adverse outcomes. Purification plays a critical role in the extraction process to yield a pure compound. To analyze the efficiency of the extraction procedure, a purified product can undergo further scrutiny using analytical techniques such as Gas Chromatography/Mass Spectrometry (GC-MS) [20], Supercritical fluid chromatography (HPLC-SFC) [21], Liquid Chromatography—Mass Spectrometry (LC-MS) [22–24], and Ultra-performance liquid chromatography–mass spectrometry (UPLC-MS) [25].

For multi-stage extraction known as sequential extraction, different solvents with varying polarities are used to isolate the target compound from different substrates [26]. The efficiency of this method is based on the polar attributes of compounds present in the raw material. Non-polar compounds such as fat, wax, pigments, fatty acids, alkane, sterols, and chlorophyll are initially extracted using a non-polar solvent (such as hexane or petroleum ether) [27]. To further solubilize the intermediate polar molecules, such as flavonoids and lipophilic chemicals, the intermediate polar solvents (such as dichloromethane and chloroform) are used [28]. Eventually, alkaloids, tannins, and other hydrophilic chemicals are among the high-polar compounds that are usually extracted using high-polar solvents such as methanol and ethanol [29]. This method is potentially effective for the isolation of α-M from mangosteen pericarps.

The efficient isolation of α-M from pericarp typically employs a three-stage sequential solvent approach, involving water, hexane, and acetonitrile. This methodology streamlines the extraction process, yielding α-M with elevated levels of purity and quantity, all without necessitating the utilization of sophisticated apparatus within the experimental framework. This investigation endeavors to formulate a direct and economical procedure for procuring α-M of superior quality, thereby obviating the requirement for specialized instrumentation or intricate

methodologies commonly associated with traditional solvent extraction techniques. The arrangement of solvents in the extraction sequence was determined based on the solvent distribution factor. Subsequently, assessments were conducted to ascertain the yield, purity, and recovery of α-M. Finally, a precipitation technique was employed to generate solid α-M powder.

## 2. Material and methods

### 2.1 Chemical and solvent

Analytical-grade organic solvents, namely hexane and acetonitrile, employed in the extraction stage were purchased from RCILabscan. Acetonitrile (HPLC-grade), used for the preparation of the mobile phase for HPLC analysis, was acquired from Supelco, Sigma Aldrich. In-house deionized water was used as a solvent for extraction and mobile phase preparation. The standard α-M (alpha-Mangostin) used in the study was sourced from Sigma Aldrich, with a declared purity exceeding or equal to 98%.

### 2.2 Preparation of mangosteen pericarp powder

Fresh Mangosteen fruits were purchased from Simummuang Market (Pathum Thani province), Thailand. The pericarps were peeled off from the stem and fruit to use as a raw material for extraction. The steps involved in preparing powder-dried mangosteen pericarps include washing, drying, grinding, sieving, packing, and storage. Mangosteen pericarps were cut into small pieces and dried at 80˚C overnight in an oven to remove the moisture in the pericarps. The dried pericarps were ground into powder using a grinder to reduce the particle size. Then the powder was sieved and separated into different fractions. to produce fine powder (< 149 μm). To avoid insect and microbiological infestations, the powder is vacuum-packed and stored at -20˚C.

### 2.3 Mass fraction of α-M in mangosteen pericarp by crosscurrent extraction

The content of α-M in Mangosteen Pericarp can be different depending on several factors such as the land area of cultivation, the weather, the irrigation, etc. To provide other researchers with the content of α-M in our raw material as a basis of our research, a series of extractions was performed. The procedure of multistage crosscurrent extraction by batch-wise operation is shown in Fig 1 to determine the mass fraction of the α-M inside the dried mangosteen powder.

$$x_0 = \Sigma \frac{E_k}{S_{k-1}} y_k \qquad (1)$$

S = mass of feed powder
$x_0$ = mass fraction of component ($\alpha$) in feed powder
E = mass of solvent
y = mass fraction of component ($\alpha$) in solution

### 2.4 Solvent distribution factor for each solvent and sequential solvent extraction

The solvent distribution factor serves as a valuable tool in the selection of solvents for extraction procedures, facilitating the efficient recovery of valuable compounds. Analyzing the

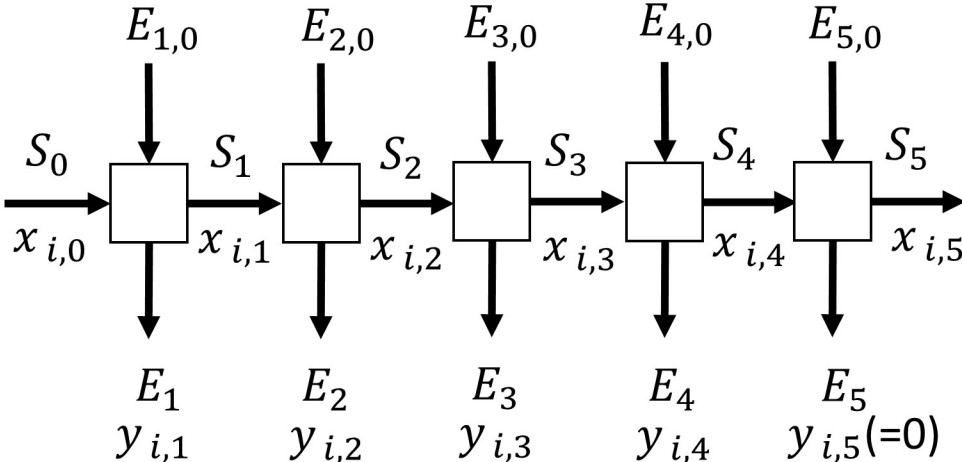

**Fig 1. Crosscurrent extraction.** In this experiment, the substrate powder (S) was mixed with solvent (E). The mass fraction of α-M in the powder and in the solution is designated as x and y, respectively. The mangosteen pericarp powder was placed in a glass tube to be extracted. At the bottom of the column, there was a valve (initially closed) that was used to collect liquid from the column by gravity. The amount of powder in the 1$^{st}$ stage is denoted as $S_0$. Then 50 mL of solvent (Acetonitrile), $E_{1,0}$, was poured into the column, providing the solid and solvent ratio of 1 g: 8.24 mL. The solvent passed through the column retention time for 10 mins at room temperature with the percolation method for each batch extraction. The solvent extract $E_1$, was removed by pipetting out with a syringe, and filter, and analyzed that extract with HPLC. The solid $S_1$ was used as feed for the 2$^{nd}$ extraction and the same amount of solvent $E_{2,0}$ was added. The process was repeated for 8 cycles. The mass fraction was calculated using Eq (1).

distribution factor's magnitude enables the formulation of potential sequences for solvent extraction aimed at removing impurities. Understanding the solvent distribution factor enhances overall process efficiency, enabling improved yields and reduced energy consumption. The solvent distribution factor (m) is calculated by the following Eq (2).

$$m = \frac{y}{x} \tag{2}$$

m = solvent distribution factor,
$y$ = mass fraction of α-M in the solution,
$x$ = mass fraction of α-M in the powder.

## 2.5 Extraction for α-M mangosteen pericarp

Different solvents with different polarities were used to extract α-M from the pericarp powder. First, water (polar solvent) was used for extraction aiming to remove impurities from the substrate with the ratio of solvent and solid of 1 g: 8.24 mL using thermo-shaker (Grant-bio, PHMT, PSC24) operating at 1000 rpm and 45°C for 10 min to facilitate mass transfer and enhance extraction efficiency. After extraction, the mixture was separated into solid and liquid layers using a centrifuge at 3000 rpm for 15 mins. The solution was withdrawn via a syringe fitted with 0.22 μm membrane filter for HPLC analysis. The residue pericarp powder was then extracted by using hexane (non-polar solvent), following the same procedure as described above. Finally, acetonitrile (intermediate polar solvent) was used to extract the treated powder. The extract solution obtained from each step was analyzed by HPLC to evaluate the % yield and % recovery of α-M. The sequential extraction procedure is shown in Fig 2.

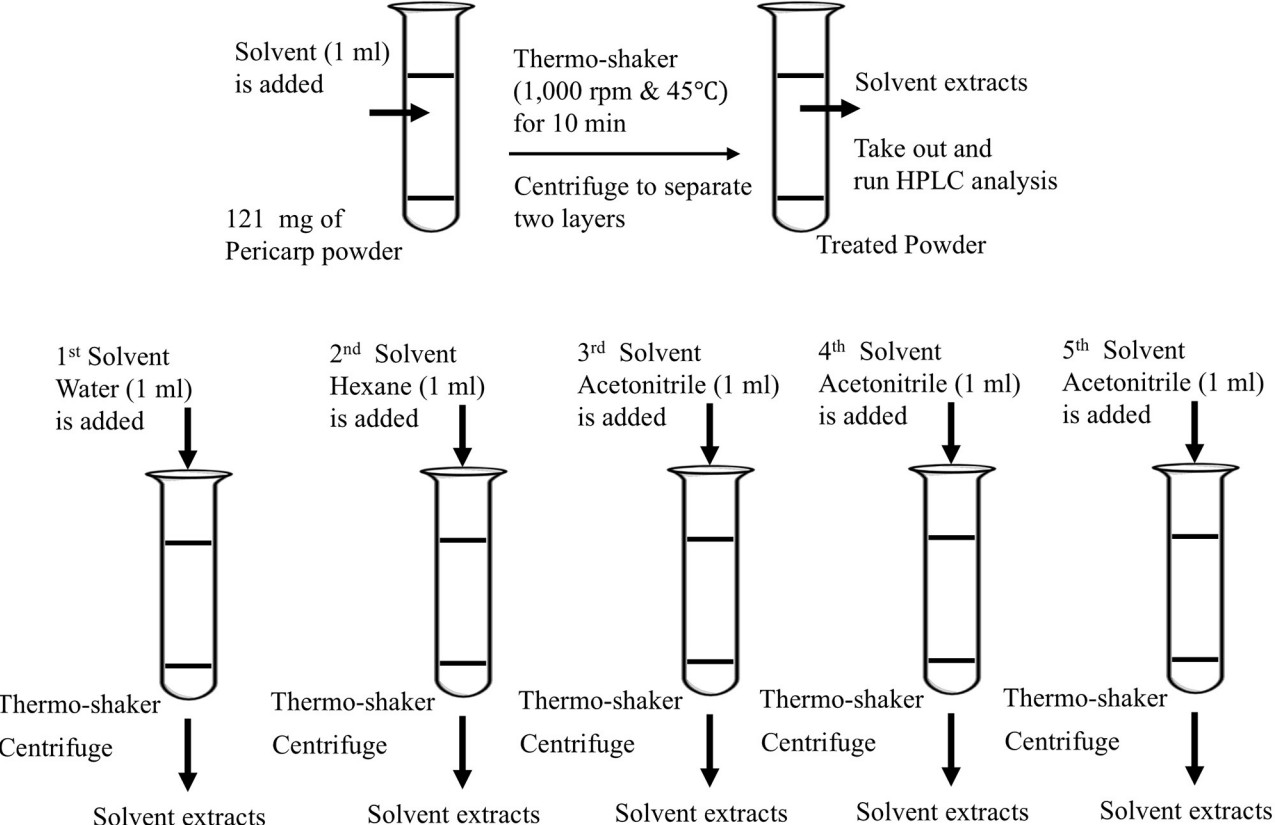

**Fig 2. Sequential extraction method.**

The yield of the α-M in the extraction solution was calculated using Eq (3).

$$Y = \frac{E\,y}{S_0} \tag{3}$$

Y = the extraction yield of α-M,
E = volume of the extract solution,
y = mass fraction of α-M in the solution,
$S_0$ = mass of the mangosteen powder.

## 2.6 HPLC analysis

The Gradient HPLC system Azura DAD P 6.1 L equipped with C18 ACE Excel 5 column with dimensions of 4.6 mm × 250 mm and a particle size of **5** μm was used for the quantification of α-mangostin at the wavelength of 320 nm. The column was maintained at 20˚C in a column compartment. The mobile phase was a mixture of HPLC-grade pure acetonitrile and distilled water. Both solvents were degassed and filtered through a 0.22 μm membrane filter prior to the analysis. The gradient elution is summarized in Table 1. The injection volume was 10 μL and the analysis spanned over 35 min.

**Table 1. Gradient conditions.**

| Time (min) | DI (%) | Acetonitrile (%) |
|---|---|---|
| initial | 15 | 85 |
| 20 | 30 | 70 |
| 22 | 10 | 90 |
| 35 | 10 | 90 |

## 2.7 Preparation of standard and calibration curve

The preparation of the HPLC standard calibration curve began by obtaining a known standard solution of $\alpha$-M with a concentration of 1.034 mg/ml. Subsequently, five different concentrations were prepared, including 0.52, 0.41, 0.31, 0.21, and 0.11 mg/ml, to establish a range of data points for the calibration curve. For each concentration, the solution was carefully measured and accurately diluted to ensure precision in the measurement. The standard solutions were then injected into the HPLC system, and the corresponding peak areas or retention times were recorded. By plotting the $\alpha$-M concentration against the peak area, the HPLC standard calibration curve was generated, providing a reliable basis for the quantification of the target analyte α-M in subsequent sample analyses. This meticulous process is essential for ensuring the accuracy and reproducibility of HPLC results for samples obtained from extraction experiments.

## 2.8 Precipitation of crude extract

Precipitation technique can selectively isolate target compounds from complex mixtures based on differences in solubility. It offers a simple, cost-effective, and environmentally friendly approach for isolating and purifying target compounds from complex mixtures. The final solution extract from the sequential solvent was mixed with the deionized (DI) water to induce precipitation. In this experiment, six different volume ratios of the extract solution with DI water were employed, specifically in the proportions of 1:0, 1:0.5, 1:1, 1:2, 1:3, and 1:6. The volume of the extract solution was constant at 1 mL. Subsequently, a precipitation period of 10 min was allotted to facilitate the separation of the solid particles from the solution. The resulting solution underwent further analysis via HPLC to determine the concentration and quantity of α-M present therein. The %recovery of α-M in this process was calculated using Eq (4).

$$\% \text{ recovery} = \frac{A_0 - A_n}{A_n} \text{ x100} \tag{4}$$

$A_0$ = Amount of α-M without water dilution (1:0)
$A_n$ = Amount of α-M with water dilution, n = volume ratio of water added (0.5,1,2,3,6)
$A_{0.5}$ = the volume ratio of the extract solution with DI water, (1:0.5)

## 3. Results and discussion

### 3.1 Determination of mass fraction of α-M inside the feed mangosteen powder

The α-M content was studied via crosscurrent extraction. The result was used to assess the % recovery of α-M. In this work, HPLC analysis serves a dual purpose in chemical analysis, encompassing not only the identification of compounds but also facilitating the quantification

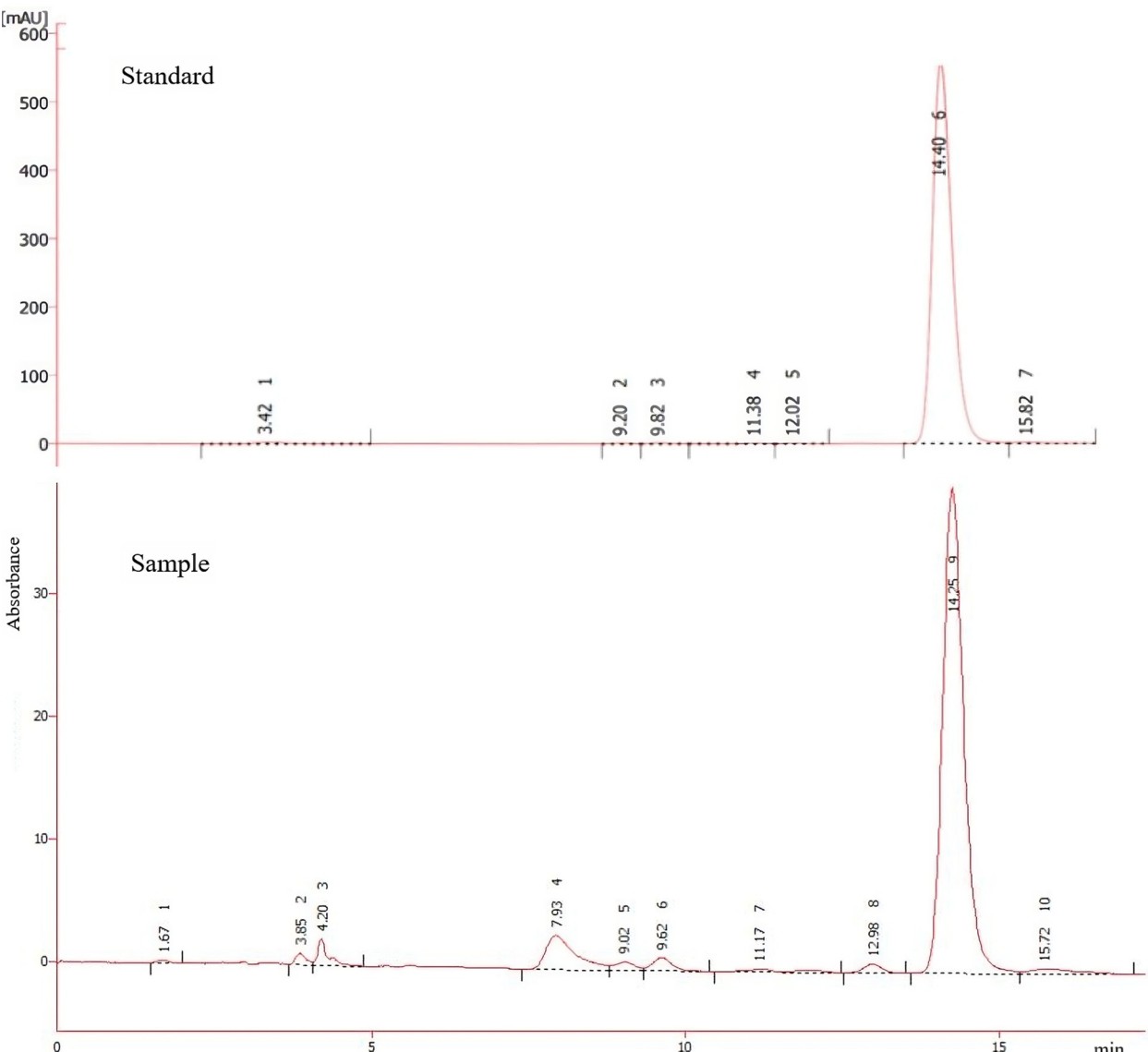

**Fig 3. Chromatogram of (a) standard of α-M (b) extract solution obtained from 1st -step percolation using acetonitrile as solvent.**

of the concentration of the target compound according to the predefined standard curve. As depicted in Fig 3, both the standard α-M and the sample (mangosteen pericarp powder dissolved in acetonitrile) exhibit a retention time of 14.40 and 14.25 min, respectively. Note that small peaks present in Fig 3B represent chemical species that were extracted simultaneously with α-M. Chromatograms obtained under different operating conditions of HPLC also resulted in similar retention times. Therefore, the highest peak in the chromatogram was suggested as α-M.

Following the derivation of concentration data from the HPLC analysis results and the standard curve, subsequent calculations may be undertaken to assess the yield, purity, quantity of α-M, and recovery rate.

A graphical representation delineating the determination of mass fraction through the utilization of the percolation method is provided in Fig 4. Apparently, the extraction yield of α-M

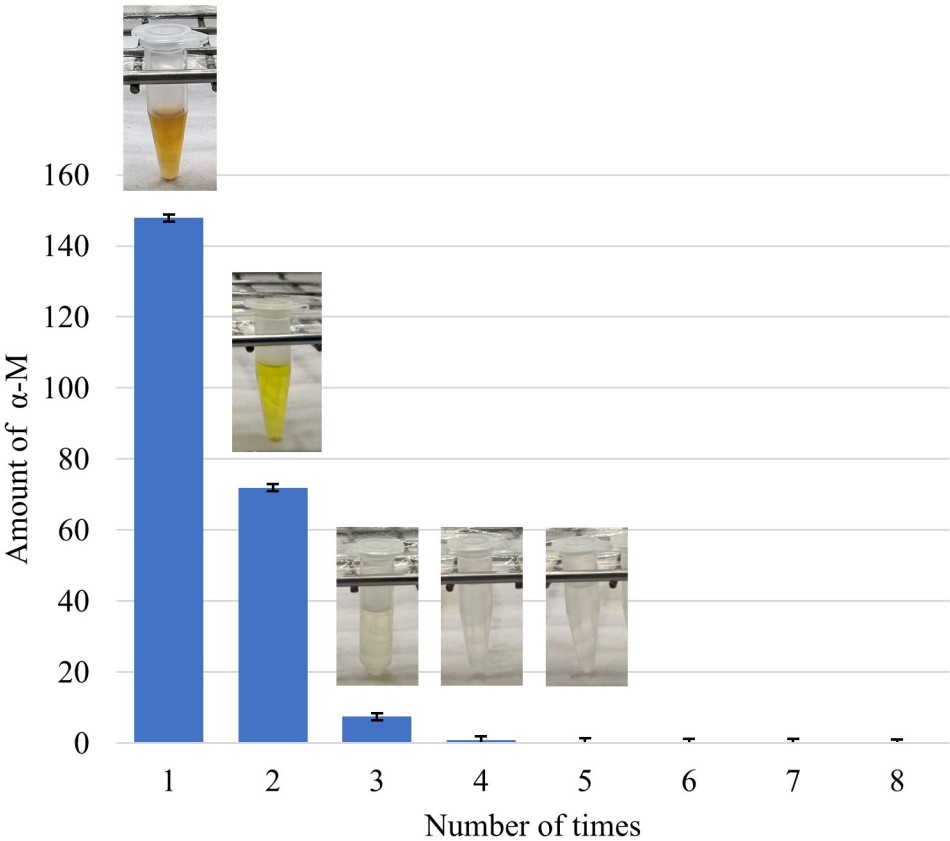

**Fig 4. Amounts of α-M obtained from multi-stage percolation.**

declined rapidly as the number of extraction cycles increased. According to this experimental procedure, the percolation method was systematically applied in successive steps. The initial step yielded a substantial quantity of α-M, approximately 147.91 mg, followed by 71.94 mg in the second step and 7.45 mg in the third step. This was also in line with the color of the extract solution as shown in the embedded pictures in Fig 4. Notably, the extract solution was colorless from the third cycle onwards. This observation indicated the complete elution of α-M at the third stage. Therefore, the first three steps were particularly effective in extracting a significant portion of α-M. The total yield of α-M was 64.2 mg/g, corresponding to the mass fraction of 0.0642.

## 3.2 Solvent distribution factor (m)

The value of the distribution factor serves as a predictive parameter for estimating the potential yield of α-M under specific extraction conditions. A high value of m generally indicates a more favorable distribution of α-M into the extracting solvent, suggesting a more efficient extraction. The m values corresponding to various solvents including water, hexane, and acetonitrile were evaluated from equilibrium extraction experiments. These values are determined through calculations involving the ratio of the mass fraction of α-M within the solution to that within the solute. The relative polarity of water, hexane, and acetonitrile is 1, 0.009, and 0.46, respectively. According to Fig 5, the acetonitrile solvent has the highest distribution factor (4.09) followed by hexane ($1.03 \times 10^{-2}$) and water ($4.5 \times 10^{-5}$). These results suggested that

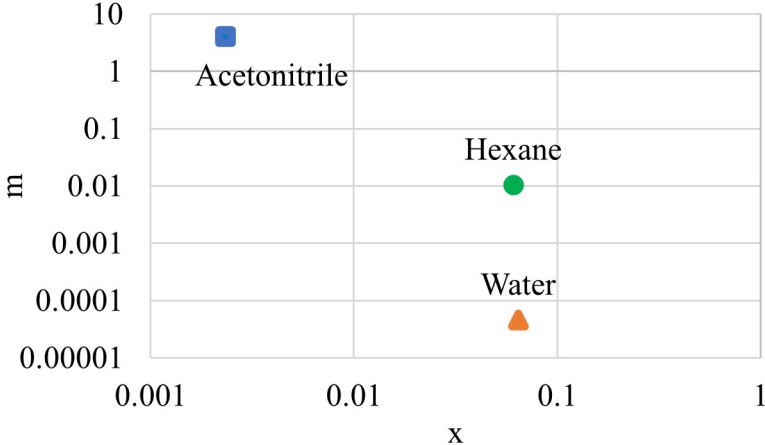

**Fig 5. Solvent distribution factor of acetonitrile, hexane, and water.**

acetonitrile (intermediate polarity) was effective for extracting α-M, while negligible amounts of α-M were extracted using hexane and water. Nevertheless, hexane and water could function as purifiers in this system targeting high-polarity and low-polarity compounds present in the mangosteen pericarp. This will be discussed in the following section on the application of sequential extraction for the isolation of α-M from mangosteen pericarp.

### 3.3 Sequential solvent extraction

The effectiveness of extracting natural products heavily relies on both the extraction solvent chosen and the properties inherent to the specific compound being targeted. The selection of extracting solvents is commonly influenced by their polarities, as outlined in reference [30]. Polarity is determined by the solubility of compounds, which, in turn, is influenced by the collective molecular properties [31]. Extracting solvents requires sufficient time to interact with the sample, allowing for the maximal transfer of α-M from the pericarp powder to the solvent. In these experiments, a period of 10 min was used to achieve the equilibrium extraction. was used, for different solvents including water, hexane, and acetonitrile, to optimize the yield of α-M. This sequential extraction involves the successive use of three solvents: water (high polarity), hexane (non-polar), and acetonitrile (intermediate polarity).

The concept of our sequential extraction was to remove the high-polarity compounds from the mangosteen pericarp powder in the first stage of extraction using water as solvent. The extract solution was analyzed by HPLC and the chromatogram is shown in Fig 6A. The retention time of polar impurities was observed within approximately 3 min. Note that none of the α-M was eluted at this stage. Then the residue from the first stage was extracted by hexane. It is effective in extracting non-polar or low-polarity compounds that were not captured by the water extraction. Conceivably, these two steps should leave the major content of α-M in the residue unaffected after the second stage of extraction. Fig 6B shows the chromatogram of the 2nd-stage extract solution. It was observed that a slight proportion of α-M was present along with the low-polarity compounds. The concentration of α-M in the 2nd-stage extract solution was 0.0217 mg/mL, corresponding to the amount of α-M of 0.3914 mg. In the 3rd-stage extraction, acetonitrile (representing an intermediate polar solvent) was employed. This solvent was chosen for its ability to dissolve a wide range of compounds with varying polarities, making it suitable for extracting components that fall between the extremes of water and hexane

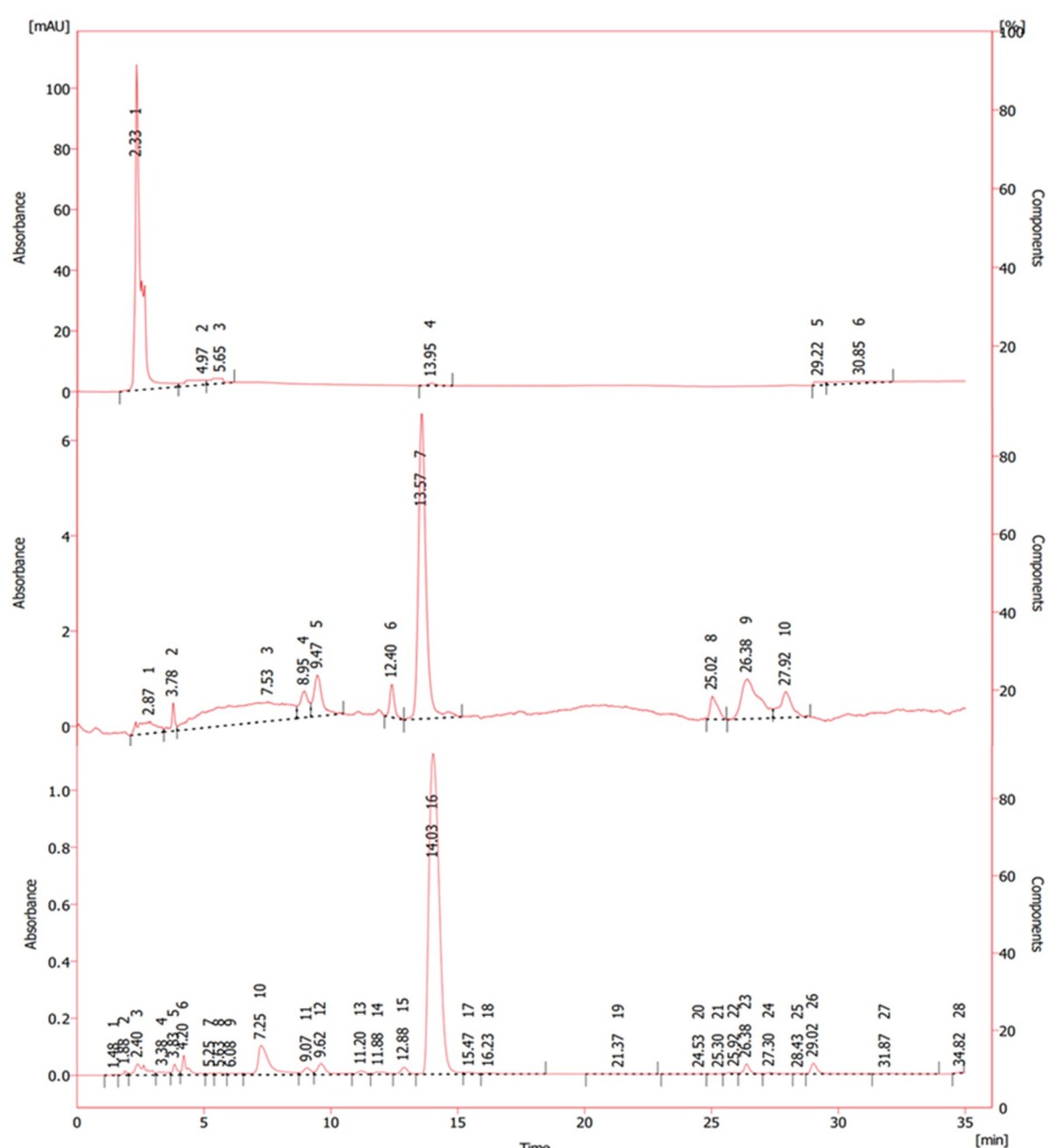

**Fig 6. Chromatogram of extract solution obtained from different stages of sequential extraction (a) water (b) water-hexane (c) water-hexane-acetonitrile.**

solubility. This step was designed to obtain the majority of α-M present in the mangosteen pericarp powder. As observed in Fig 6C, the peak area of α-M is approximately 180 times larger than that of the 2nd-stage extract. In other words, most of α-M was collected in this final step of the sequential extraction. The analysis revealed that the concentration of α-M was 5.2196 mg/mL, which is much higher than that of the 2nd-stage extract solution. The amount of α-M in the 3rd-stage extract solution was 114.8307 mg. It means that the loss of α-M due to

**Table 2. Yield using sequential solvent.**

| Solvent | Yield (wt/wt) | Recovery (%) |
|---|---|---|
| Water | 0.01 ± 0.001 mg/g | - |
| Water-Hexane | 0.13 ± 0.001 mg/g | - |
| Water-Hexane-Acetonitrile | 28.94 ± 1.45 mg/g | 45.22 ± 0.17% |
| Water-Hexane-Acetonitrile-Acetonitrile | 39.71 ± 2.41mg/g | 62.05 ± 1.61% |
| Water-Hexane-Acetonitrile-Acetonitrile-Acetonitrile | 46.75 ± 2.57 mg/g | 73.05 ± 0.53% |

the extraction by hexane in the 2$^{nd}$ step was 0.3409%. Hence, the sequence of solvent extraction as water-hexane-acetonitrile was effective for the isolation of α-M from mangosteen pericarp.

Generally, the extraction efficiency or the yield of α-M can be enhanced by increasing the number of stages. To demonstrate this effect, another sequential extraction experiment was performed by adding two stages of extraction using acetonitrile. Results are shown in Table 2. It was found that the increase in the recovery in each step was significant, suggesting that additional stages were necessary to obtain a higher yield of α-M. This was different from the percolation results in Fig 4, in which an exponential decay was observed for the yield of α-M as the number of extraction cycles increased. This was attributed to the gradual mixing of solvents (hexane and acetonitrile) in the extraction system. As the aliquot of acetonitrile was added to the extraction column, a certain portion of hexane (previously retained inside the bed of solid residue) interacted with the new solvent. Therefore, the extraction efficiency of the 3$^{rd}$ stage of the sequential extraction was significantly lower than that of the percolation experiment. Since there was no mechanical agitation in this system, the removal of hexane was based on diffusion. This also affected the following steps, as observed by the significant increase in the yield of α-M. In 5-step sequential solvent extraction, the first two steps aimed for the removal of impurities followed by the main extraction in the last three steps using acetonitrile as the solvent, the accumulated yield was 46.75 mg/g.

Furthermore, the recovery rate was assessed according to Eq 4. Applying 3-stage solvent extraction resulted in approximately 45.22% recovery. This means that about half of α-M was retained within the solid residue. Additional stages of extraction would be required to enhance the %recovery. For instance, upon an extra stage of extraction using acetonitrile improved the recovery to 62.05%. The final %recovery achieved through our 5-stage sequential solvent extraction was 73.05%, calculated based on the accumulated mass of α-M extracted and the initial mass of α-M present in the mangosteen pericarp powder. Although the amount of solvent used for this process increases with increasing the number of (additional) extraction stages, a recovery of solvent could be implemented using a simple rotary evaporator. The results obtained using fresh solvent were similar to those obtained using recycled solvent. Hence, the cost of extraction can be greatly reduced.

To determine the purity of α-M, the solution product obtained from the 5-step sequential extraction was dried in a convective oven overnight. A small amount of the dried powder was redissolved in a mixture of acetonitrile and water (75:25 by volume). The chromatogram obtained from HPLC analysis was compared with the standard curve, and the purity was calculated by Eq 5. It was found that the purity of α-M was 67.9%. While a powder with 67.9% purity of alpha-mangostin may not be as concentrated as pure alpha-mangostin extract, it still retains significant bioactivity and can be used for various purposes. Note that higher purity offers better control of the content of bioactive compounds in the product. Some potential uses include dietary supplements, anti-inflammatory agents, antimicrobial agents, functional

food ingredients, and cosmetic ingredients.

$$\% \text{ purity} = \frac{mass\ of\ \alpha - M\ in\ the\ solution}{mass\ of\ dried\ powder}\ \text{x}100 \qquad (5)$$

### 3.4 Recovery of α-M via precipitation

In order to collect the crude extract in the solid form, the extract solution obtained from sequential extraction was further processed. In general, rotary evaporation or speed vacuuming for solvent removal from the product can be applied. However, this requires additional equipment and energy cost of production. In this work, an alternative method of precipitation offers a simplified and facile approach. This method involved precipitation induced by the addition of DI water to the extract solution. The efficiency of this water-precipitation technique stems from the partial miscibility of water and acetonitrile, owing to the nature of their molecular interactions. In principle, DI water is readily mixed with acetonitrile due to different molecular interactions such as hydrogen bonding, dipole-dipole interaction, dispersion forces, and solvation. Among these interactions, hydrogen bonding is typically the strongest interaction in a solution of water and acetonitrile. The prevalence of hydrogen bonding in water underscores its unique properties, including high solubility for certain polar substances [32]. This interaction, modulated by parameters such as temperature, pressure, and chemical composition of the mixture, underpins their partial miscibility. The advantageous properties arising from partial miscibility find application in diverse chemical reactions and separation processes, prompting researchers to leverage these solvents for their distinctive characteristics across various domains [33]. As a result, the amount of acetonitrile molecules necessary to dissolve the crude extract was affected. Due to the above factors, the interaction between acetonitrile and α-M (as well as other compounds present in the crude extract) was weakened as most of the acetonitrile molecules were interacting with water. and starting to precipitate the. Hence, the precipitation of powder α-M crude extract was observed. This part of our study was designed to explore the ratio of DI water and extract solution that could recover the crude extract completely. This ratio was directly related to the number of molecules of each species; therefore, the concentration of extract solution of 14.01 mg/mL was kept constant for this set of experiments. Results, as shown in Fig 7, indicated that the recovery of α-M was strongly affected by the volume ratio of the α-M solution relative to deionized (DI) water. Upon applying the ratio of 1:6, the recovery of α-M from the extract solution was nearly complete. This was confirmed by a trace amount of α-M found in the final solution.

### 3.5 Comparison of extraction performance

Based on the literature review, the performance of conventional extraction of mangosteen pericarp was compared with our sequential extraction as presented in Table 3. Maceration, a widely used approach, yields 1.19 mg/g of xanthones from mangosteen pericarp when macerated with 95% ethanol for seven days [13]. Tran et al. (2021) reported a xanthone yield of 5.626 mg/g from mangosteen pericarp after 96 h of ethyl acetate maceration of peel powder [34]. It is worth noting that ethyl acetate is less polar than ethanol due to its ester functional group. Microwave-assisted extraction using ethyl acetate as a green solvent achieved the highest α-mangostin (α-M) yield at the optimal conditions of 189.20 W power, 72.40% (v/v) ethyl acetate, and the extraction time of 3.16 min. The α-M content in the extract reached 121.01 mg/g dry matter (DM) [15]. However, the sophisticated equipment requirements, energy consumption, and maintenance associated with this technique represent significant drawbacks

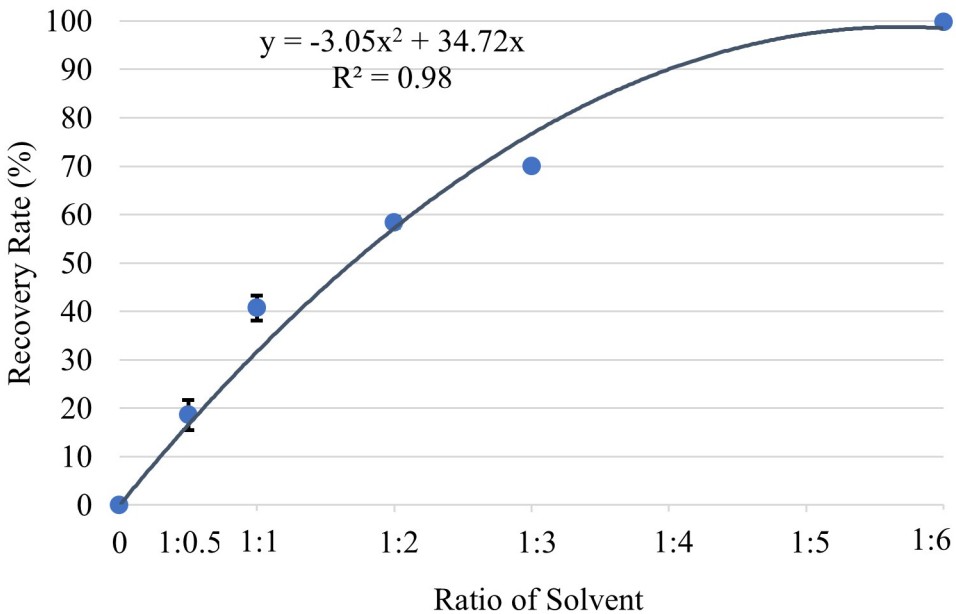

$$y = -3.05x^2 + 34.72x$$
$$R^2 = 0.98$$

**Fig 7. Recovery of crude extract via precipitation.**

compared to other methods. Percolation, a continuous method, yields 12.71% w/w α-M when using a percolator with 95% ethanol. Soxhlet extraction, a traditional technique, suffers from drawbacks such as large solvent volumes and extended extraction periods [35]. Soxhlet extraction yielded a crude extract comprising 26.60% of the dry weight, with α-mangostin constituting 13.51% (w/w of the crude extract), achieved using a concentration of 50% ethanol [35]. Conventional methods such as maceration, soxhlet extraction, and heat reflux are time-consuming while offering low product recovery. Therefore, our three-stage or five-stage percolation technique can be an efficient alternative to recover α-M from mangosteen pericarp. The process is simple, and processing can be completed in a short time. Note that the solvent used in the extraction process can be recovered. Hence, the cost of solvent can be greatly reduced. Note that, unlike conventional extraction processes, our sequential extraction also yielded 46.75 mg/g α-M from mangosteen pericarp with a purity of α-M of 67.9%. Achieving high

**Table 3. Extraction performance of different techniques.**

| Extraction Method | Conditions | Active Compounds Yield |
|---|---|---|
| Maceration [13] | macerated with 95% ethanol for 7 days | yields 1.19 mg/g of xanthones from mangosteen pericarp |
| Maceration [34] | 500 g of peel powder macerated with 1L of ethyl acetate for 96 h | yields 5.626 mg/g of xanthones from mangosteen pericarp |
| Microwave [15] | Ethyl acetate as a green solvent followed by dichloromethane, ethanol, and water exhibited the highest α-M at 3.16 min, 189.20 W, and 72.40% (v/v). | α-M concentration in mangosteen pericarp of 121.01 mg/g dry matter (DM) |
| Percolation [35] | using a percolator with 95% ethanol | 12.71% w/w α-M |
| Soxhlet extraction [35] | using a concentration of 50% ethanol | crude extract (26.60% dry weight), α-mangostin (13.51%, w/w of crude extract) |
| Proposed Method | Sequential Solvent Extraction (Water, Hexane, Acetonitrile) | yields 46.75 mg/g α-M with 67.9% purity from mangosteen pericarp |

purity ensures that the extracted compound maintains its biological activity, allowing for accurate evaluation of its potential health-promoting properties.

## 4. Conclusion

Alpha-mangostin, a valuable bioactive compound, was successfully extracted from mangosteen pericarp through an optimized sequential solvent extraction process. By systematically employing water, hexane, and acetonitrile, we achieved a significant enhancement in both yield (46.75 mg/g) and purity (67.9%) of alpha-mangostin compared to traditional methods. The incorporation of additional acetonitrile extraction stages proved instrumental in improving recovery rates and eliminating impurities. The precipitation method demonstrated efficacy in recovering the extracted product. This study provides a robust foundation for the development of a scalable process for the production of high-purity alpha-mangostin from mangosteen pericarp.

## Supporting information

**S1 File.**
(DOCX)

## Author Contributions

**Conceptualization:** Moh Moh Han, Preuk Tangpromphan, Amaraporn Kaewchada, Attasak Jaree.

**Formal analysis:** Moh Moh Han, Attasak Jaree.

**Funding acquisition:** Attasak Jaree.

**Investigation:** Moh Moh Han.

**Methodology:** Moh Moh Han, Preuk Tangpromphan, Amaraporn Kaewchada, Attasak Jaree.

**Project administration:** Amaraporn Kaewchada.

**Resources:** Preuk Tangpromphan, Amaraporn Kaewchada, Attasak Jaree.

**Supervision:** Attasak Jaree.

**Visualization:** Moh Moh Han.

**Writing – original draft:** Moh Moh Han, Amaraporn Kaewchada, Attasak Jaree.

**Writing – review & editing:** Attasak Jaree.

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
