## [Decision Letter · Decision Letter 0]

3 Jul 2024

PONE-D-24-07293Recovery and Partial Isolation of ⍺-Mangostin from Mangosteen Pericarps (Thailand) via Sequential Extraction and PrecipitationPLOS ONE

Dear Dr. Jaree,

Thank you for submitting your manuscript to PLOS ONE. After careful consideration, we feel that it has merit but does not fully meet PLOS ONE’s publication criteria as it currently stands. Therefore, we invite you to submit a revised version of the manuscript that addresses the points raised during the review process.

We look forward to receiving your revised manuscript.

Kind regards,

Fahrul Nurkolis

Academic Editor

PLOS ONE

Journal Requirements:

2. Thank you for stating the following financial disclosure: "Faculty of Engineering, Kasetsart University".

3. We note that your Data Availability Statement is currently as follows: "All relevant data are within the manuscript and its Supporting Information files."

Additional Editor Comments:

Your manuscript has now been reviewed by experts in the field. Please revise the manuscript according to the referees' comments.

Reviewers' comments:

Reviewer's Responses to Questions

**Comments to the Author**

1. Is the manuscript technically sound, and do the data support the conclusions?

Reviewer #1: Yes

Reviewer #2: Partly

2. Has the statistical analysis been performed appropriately and rigorously? 

Reviewer #1: Yes

Reviewer #2: I Don't Know

3. Have the authors made all data underlying the findings in their manuscript fully available?

Reviewer #1: Yes

Reviewer #2: Yes

4. Is the manuscript presented in an intelligible fashion and written in standard English?

Reviewer #1: Yes

Reviewer #2: Yes

5. Review Comments to the Author

Reviewer #1: Manuscript describes isolation of alpha-mangostin from the pericarps in Thailand. The manuscript is very well written with good experimental section. Thus, I would suggest this interesting paper for publication in current form in this journal.

Reviewer #2: Thank you for the opportunity given to make corrections to the article with the title "Recovery and Partial Isolation of ⍺-Mangostin from Mangosteen Pericarps (Thailand) via Sequential Extraction and Precipitation". Several aspects need to be addressed in this article from background to conclusion. The following points need to be addressed in this article:

1. Introduction

- The explanation of mangostin on the first page is too much, so I suggest combining paragraphs two, three, four, and five into a single short paragraph.

- It is best not to include images in the background

- There are no citations showing the percentages of α-mangostin and gamma-mangostin. Should be added

- The introduction prepared in this text contains many definitions and meanings. This section should be made more concise by directly explaining the essence of the problem being discussed. Definitions and understandings that are excessive make the reader lose focus on the main problem being conveyed. It's best to just describe the main problem clearly and concisely to attract attention and make it easier for readers to understand.

- Currently, the paragraph describes extraction, purification, and then back to extraction. This order needs to be rearranged because It is confusing. It is best to first explain the extraction process in full, and then continue with an explanation of purification. With a more logical and systematic sequence, readers can more easily understand the flow and stages.

2. Materials & methods

- For the factors you mentioned regarding the content of the alpha mangostin compound in the pericarp, what function does this information have in your research? Do you also perform mangostin extraction on the pericarp, which is influenced by these factors? If yes, it also includes data regarding treatment. If there is any from other research, include data from that research

- The formula used in this manuscript was inconsistent. If you want to use a list of information, you should place it below the formula. However, if you want to use paragraphs, use them consistently to explain the formula. Consistency in writing format will help readers to understand formulas and information more easily and clearly.

3. Results and Discussion

- You stated that the mangostin taken was from the Bangkok market, but why do you mention a different source here? Please explain

- The length of an abbreviation must be stated at the beginning of the abbreviation so that readers are not confused

- I see that you use reference journals 13 and 33 as the sources of the large number of alpha mangostin compounds that were successfully extracted. These two sources are research conducted in 2010, and there are no years. Therefore, research sources that go back at least 5 years are needed so that the novelty of the research is clearer

4. Conclusion

- The conclusions presented by the researcher need to be shortened again

5. Bibliography

- 94% of the references used in this article came from articles published more than five years ago. This value makes the novelty of this research questionable. Therefore, researchers must look for other reference sources to replace the references that have been used

- Some references do not have a year, such as reference numbers 33 and 34. This should be corrected again

This research offers an extraction method that does not involve too many tools and yields good results. However, this article must be revised to make it suitable for publication. Based on the suggestions and input I have provided, I decided that this article requires major revisions.

6. PLOS authors have the option to publish the peer review history of their article (what does this mean?). If published, this will include your full peer review and any attached files.

Reviewer #1: No

Reviewer #2: No

---

## [Author Response · Author response to Decision Letter 0]

3 Aug 2024

We sincerely appreciate the reviewers for their valuable comments and suggestions. We have carefully addressed all points raised and incorporated the necessary revisions into the manuscript. A detailed response outlining the specific changes made can be found in the attached document, “Response to Reviewer.docx”.

---

## [Editor Report · Decision Letter 1]

1 Sep 2024

Recovery and Partial Isolation of ⍺-Mangostin from Mangosteen Pericarps (Thailand) via Sequential Extraction and Precipitation

PONE-D-24-07293R1

Dear Dr. Jaree,

We’re pleased to inform you that your manuscript has been judged scientifically suitable for publication and will be formally accepted for publication once it meets all outstanding technical requirements.

Kind regards,

Fahrul Nurkolis

Academic Editor

PLOS ONE
---

## [Editor Report · Acceptance letter]

16 Oct 2024

PONE-D-24-07293R1 

PLOS ONE

Dear Dr. Jaree, 

I'm pleased to inform you that your manuscript has been deemed suitable for publication in PLOS ONE. Congratulations! Your manuscript is now being handed over to our production team.

Kind regards, 

on behalf of

Dr. Fahrul Nurkolis 

Academic Editor

PLOS ONE